# Form-Stable Phase Change Materials Based on SEBS and Paraffin: Influence of Molecular Parameters of Styrene-b-(Ethylene-*co*-Butylene)-b-Styrene on Shape Stability and Retention Behavior

**DOI:** 10.3390/ma13153285

**Published:** 2020-07-23

**Authors:** Ralf Rickert, Roland Klein, Frank Schönberger

**Affiliations:** Fraunhofer Institute for Structural Durability and System Reliability LBF, Schlossgartenstr. 6, 64289 Darmstadt, Germany; ralf.rickert@lbf.fraunhofer.de (R.R.); roland.klein@lbf.fraunhofer.de (R.K.)

**Keywords:** phase change material, SEBS, paraffin, form-stable phase change material, phase transition, shape stability mechanism, leakage, thermal energy storage, hexadecane

## Abstract

In this work, the influence of molecular parameters of styrene-b-(ethylene-*co*-butylene)-b-styrene (SEBS) triblock copolymer as matrix material in form-stable phase change material (FSPCM) on the thermo-mechanical properties and leakage behavior are studied. Various SEBS grades differing in their molecular weight, styrene content, and ethylene/butylene ratio are used as supporting matrix in composites with 90 wt.% paraffin. Thermo-mechanical properties are determined by rheological measurements. The results show phase transitions temperatures from solid to hard gel, hard gel to soft gel, and soft gel to gel fluid. Paraffin leakage in FSPCM is analyzed by mass loss over time in an oven at 60 °C. Differential scanning calorimetry (DSC) and thermogravimetric analysis (TGA) are applied to determine the thermal energy storage capacity. Finally, the molecular weight and the styrene content are combined to the molecular weight of styrene block which is identified as the authoritative parameter for the thermo-mechanical properties of the SEBS/PCM composite.

## 1. Introduction

The continuous increase of greenhouse gas emissions into atmosphere are the main cause of global warming and have become a serious environmental problem worldwide. Great efforts have been made to reduce the use of fossil fuel and increase the utilization of renewable energy sources. Thermal energy storage (TES) is one of the most promising ways to enhance efficiency in energy saving to use available heat sources efficiently. There are three common principles for TES, like sensible energy, latent heat on fusion and freezing of phase change materials (PCMs), or chemical reactions of some products. Due to their high storage density and small temperature ranges for melting and freezing, PCMs attract great attention [1,2,3]. PCMs are well studied and are available in a wide range of application temperatures and melting enthalpies. They can be divided into two classes: (i) inorganic PCMs, including salt hydrates, salts, metals and alloys; and (ii) organic PCMs, including paraffins, fatty acids, fatty esters, and sugar alcohols. Among the classes, the paraffins are the most promising PCM, because of wide range phase change temperature, large melting and freezing enthalpy, minor super-cooling, low density, good thermal and chemical stability, and self-nucleating behavior [4,5,6]. However, paraffin also has disadvantages like low thermal conductivity and high inflammability. In addition, leakage may occur from the liquid phase if not properly enclosed [7,8]. PCM leakage can be reduced or even avoided by two common approaches: One is encapsulating the PCM, either on macroscopic or microscopic scale, the other one is to form-stabilize the PCM in a polymeric matrix. Macro encapsulation leads to insufficient heat transfer because of the low thermal conductivity of the paraffin. Micro encapsulated PCM mostly consists of PCM as the core and of a polymeric shell made of, for example, melamine formaldehyde resin (MF), polymethylmethacrylate (PMMA), or polystyrene (PS) with an average particle size from 120 nm to 100 µm with an encapsulation ratio up to 88% [9,10,11,12]. However, the higher the PCM content, the lower the shell thickness is and thus the lower the mechanical properties are [10,13,14,15]. An alternative to reduce the leakage of paraffin in TES is the use of so called form or shape stabilized PCMs in which the PCM is blended with a polymer matrix like polyethylene (PE), polypropylene (PP), polymethylmethacrylate (PMMA), or triblock copolymers like styrene-b-butadiene-b-styrene (SBS) or its hydrogenated form, styrene-b-ethylene-butylene-b-styrene (SEBS). When paraffin is used as the PCM, polyolefins like PP and PE are often used as the supporting materials and filled with a maximum of up to 77 wt.% PCM [3,11,16,17]. In comparison, thermoplastic elastomers like SBS or SEBS can uptake 80 wt.% paraffin in case of SBS, and 90 wt.% in case of SEBS, respectively [8,18]. This is because the EB midblocks in SEBS are chemically more similar to paraffin than the polybutadiene segments in SBS. The hydrogenated double bonds in the polybutadiene segments make SEBS chemically more inert compared to SBS. The structure of SEBS paraffin gel is schematically illustrated in Figure 1. The two different blocks, styrene as hard domain and ethylene butylene as soft domain, are phase separated. Glassy styrene domains stabilize the gel and act as physical cross-links, whereas the ethylene butylene domain preferentially contains paraffin as seen in Figure 1 [19,20,21]. So far, thermo-mechanical properties of SEBS paraffin gels have been studied in dependence of the paraffin/SEBS ratio for a given SEBS grade, but not in dependence of structural parameters of the SEBS. An increasing paraffin content leads to lower thermo-mechanical properties. Previous studies on SEBS/paraffin gels show a maximum paraffin level of up to 90 wt.% without any obvious leakage [8]. In order to gain a deeper understanding of the influence of molecular parameters of the matrix material on the phase transition temperatures of the FSPCMs, the dependency of molecular weight, styrene content, and E/B ratio is investigated systematically. Leakage behavior over time is also studied in this work by storing samples at higher temperatures in an oven.

## 2. Materials and Methods

### 2.1. Materials

Hexadecane was purchased from Rubitherm GmbH (Berlin, Germany). Different SEBS triblock copolymers were provided from Kraton Polymers GmbH (Frankfurt am Main, Germany) and Marubeni Europe (Duesseldorf, Germany).

### 2.2. Sample Preparation

SEBS/paraffin FSPCMs were prepared by the following procedure: 45 g of paraffin and 5 g of SEBS were weighed in a 500 mL three-neck round-bottom flask equipped with a reflux condenser and a mechanical stirrer. The mixture was heated to 140 °C under continuous stirring at 200 rpm. After 2 h at 140 °C, the mixture was allowed to cool down to room temperature. The gel was taken out and portioned into a quadruple cavity and hot pressed to discs with a diameter of 35 mm and a thickness of 1.5 mm. The press was closed at room temperature, heated to 160 °C, held at 160 °C for 5 min, and cooled to temperature at around 12 °C to demold the FSPCM at solid state of the PCM.

### 2.3. Methods

#### 2.3.1. Size Exclusion Chromatography

Size exclusion chromatography (SEC) was performed with high resolution styrene divinyl benzene (SDV) column set from PSS (SDV 10^3^, SDV 10^5^, SDV 10^6^) at 30 °C. Calibration was carried out using PS standards. For the SEC experiments, a system composed of a 1260 IsoPump—G1310B (Agilent Technologies Deutschland GmbH, Waldbronn, Germany), a 1260 VW-detector—G1314F—at 254 nm, and a 1260 RI-detector—G1362A—at 30 °C. For data acquisition and evaluation, PSS WinGPC^®^ UniChrom 8.2 was used. An injection volume of 100 μL, a sample concentration of about 2 mg/mL, a column temperature of 30 °C and a tetrahydrofuran (THF) flow rate of 1 mL/min was used.

#### 2.3.2. Nuclear Magnetic Resonance (NMR) Spectroscopy

1. H NMR spectra of dissolved samples were recorded at a BRUKER NanoBay 300 (7.05 T, Rheinstetten, Germany) using a 5 mm BBFO probe. The following parameters were used: relaxation delay of 5 s, number of scans 16, temperature at 26 °C, and deuterated chloroform (CDCl_3_) as solvent. Trimethylsilane (TMS) was used as internal standard.

#### 2.3.3. Differential Scanning Calorimetry

A Mettler-Toledo DSC822e/c (Mettler Toledo, Columbus, OH, USA) was used to evaluate the thermal properties of the SEBS paraffin gels. Heat capacity measurements of samples, punched out of sheets with a diameter of 4 mm and a thickness of 1.5 mm, were carried out in a temperature range from –20 to 50 °C with heating and cooling rates of 5 K/min. Two cycles were measured to erase thermal history. Both the first cooling and the second heating phase were used for evaluation of melting point *T_m_*, latent heat enthalpy (Δ*H*) and on-/offset of enthalpy peak.

#### 2.3.4. Thermogravimetric Analysis

The thermal stability of the SEBS paraffin gel and the amount of PCM in the sample were characterized using a METTLER TOLEDO TGA/DSC STAR System thermogravimetric analyzer (Mettler Toledo, Columbus, OH, USA) combined instrument. Each sample with a same size as for DSC were heated from 35 to 600 °C with a heating rate of 10 K/min under nitrogen atmosphere with a flow rate of 40 mL/min. In order to minimize the influence of mass, the samples were prepared in a small mass range from 13.1 to 13.6 mg.

#### 2.3.5. Rheology Measurement

A parallel plate rheometer from Haake (Mars Modular Adcaneced Theometer System, Thermo Fisher Scientific Inc, Waltham, MA, USA) was used to investigate the shape stability of the SEBS/paraffin FSPCMs. Samples with a diameter of 35 mm and a thickness of 1.5 mm were used for testing. Dynamic heat ramp tests from 15 to 200 °C were performed at a frequency of 1 rad/s with a heating rate of 2 °C/min. A single shear strain of 1.0% was applied for all measurements. The gap thickness was 1.2 mm to ensure a permanent compressive force to avoid delamination from the plate.

#### 2.3.6. Leakage Test

Paraffin leakage was tested by placing the sample (Ø 35 mm; thickness 1.5 mm) between a stack of 10 layers of filter paper from VWR International (Darmstadt, Germany) (type 434 and a diameter of 90 mm). The sandwich was placed in a preheated oven at 60 °C. Mass loss of the samples was detected by a Mettler Toledo AS105 balance (Mettler Toledo, Columbus, OH, USA). The sample was removed from the filter paper for each measurement and replaced afterwards.

## 3. Results and Discussion

### 3.1. Chemical Analysis of SEBS Types

SEBS is a triblock copolymer with two hard styrene blocks at both ends of the polymer chain and a soft ethylene/butylene copolymer in the middle as seen in Scheme 1. Various SEBS types were selected on the basis of the following molecular parameters: molecular weight, styrene content, and E/B ratio. A set of at least two SEBS types for each parameter was selected to determine the influence of the individual molecular parameters (cf. Table 1, dependency of molecular weight; Table 2, dependency of styrene content; Table 3, dependency of ethylene/butylene ratio). SEC and NMR were used for measure all SEBS materials. The samples were denominated as SEBS-molecular weight-styrene content-ethylene content in E/B phase-paraffin. For example, SEBS-345-28-64-hex has a molecular weight of 345 kg/mol, styrene content of 28 wt.%, 64% ethylene content in E/B phase and hexadecane as PCM. All further evaluations consist of samples containing 90 wt.% hexadecane and a selected SEBS type from Table 1, Table 2, or Table 3.

### 3.2. Thermal Properties of SEBS/Paraffin FSPCMs

The TGA curves of hexadecane and FSPCMs are shown in Figure 2. The temperature evaluated at 1 wt.% and 85 wt.% mass loss are summarized in Table 4. Hexadecane fully evaporates in one single step and no residue is observed at temperatures above 212 °C. In contrast, all FSPCMs show two steps of weight loss. The first step in the temperature range of 212to 221 °C can be attributed to hexadecane evaporation and corresponds to a mass loss of about 90 wt.%. It is worth mentioning that hexadecane evaporates slower from the FSPCM samples than from the pure hexadecane sample, indicating additional interaction between the SEBS and hexadecane. In the second step, the remaining SEBS degrades residue-free in the temperature range of 459 to 466 °C. T_99%_ represents the temperature at 1 wt.% mass loss, and is at about 110 °C for all the FSPCMs. No correlation between hexadecane evaporation and the molecular weight, styrene content or E/B ratio in SEBS is observed for T_99%_. However, slight changes can be determined at higher loss of mass. The temperature at which 15% of the initial mass remain (T_15%_) show the lowest value for pure hexadecane with 206 °C. Comparing T_15%_ for the composites differing in M_w_ shown in Table 1 decreases slightly from 220 °C to 212 °C. A shift to lower temperatures with decreasing molecular weight of the SEBS matrix is obvious. However, comparing the T_15%_ of composites in Table 2, there is no difference even with a varying styrene content up to 11 wt.% points. A variation of E/B as parameter show also no influence in T_15%_ is seen by comparing SEBS-085-18-52-hex and SEBS-087-18-27-hex in Table 3. The results of TGA measurements suggest a thermal stability up to 113 °C and a slight influence of molecular weight of SEBS matrix on evaporation of hexadecane. A significant influence of styrene content and E/B ratio is not obvious.

The heating and cooling DSC characteristics of hexadecane and SEBS/hexadecane composites are shown in Figure 3 and summarized in Table 4. Crystallization of hexadecane occurs in a narrow temperature range of about 3.9 K with an enthalpy of 235 J/g. All melting enthalpies of the FSPCM are between 187 and 194 J/g. It is obviously that the theoretical enthalpy value of about 211 J/g is not reached indicating that SEBS affects the crystallization of hexadecane [22]. The results show no significant influence of the molecular weight, styrene content, or E/B ratio in SEBS on the enthalpy of the SEBS/hexadecane composites. However, there is an obvious influence of the width of melting and crystallization range. The onset temperature of crystallization T_c1_ of all composites is between 14.2 and 15.6 °C and can be assumed to be equal. The end temperature of crystallization T_c2_, however, is not equal and differs T_c1_. It is visible, that the sample SEBS-345-28-64-hex has a wider temperature range of phase change (∆T_c_) than sample SEBS-118-12-26-hex. Obviously, the temperature range of phase change (∆T_c_) broadens according to a certain pattern, the underlying effect will be addressed in Section 3.3.

### 3.3. Rheology

The highest service temperature for shape stability of FSPCMs is of vital importance for technical use of SEBS as matrix material for paraffin as TES. Rheological measurements were applied to correlate the three molecular parameters of SEBS (molecular weight, styrene content, and E/B ratio) onto the viscoelastic behaviors of the FSPCMs. Paraffin with a chain-length of C_16_ was used for each composition. The phase transition behavior is characterized by analyzing the temperature-dependent storage modulus (G’) and loss modulus (G’’). While the storage modulus is a measure of the strength and a measure for the resistance of the FSPCM under mechanical stress, the loss modulus describes the flow behavior of the material under stress. [19]. The temperature-dependent behavior of G’ and G’’ of SEBS/hexadecane and the relative amount of liquid styrene are shown in Figure 4 as example of SEBS-138-31-35-hex. G’ and G’’ ascribe four typical thermo-mechanical states of the SEBS/hexadecan composite, solid state, hard gel, soft gel, and fluid state [8,19]. The first transition temperature corresponds to the melting point of paraffin *T*_m_, followed by the hard- to soft-gel transition temperature *T_1_* at the local maximum of G’’ (start of stronger decrease of G’). Finally, the gel–sol transition temperature *T*_2_ is defined by the intersection of G’ and G’’ [19,23]. The evaluated temperatures *T*_1_ and *T*_2_, determined with a temperature gradient of 2 °C/min are summarized in Table 5. The transition temperatures *T*_1_ and *T*_2_ for composition SEBS-345-28-64-hex could not be determined at 2 °C/min because of evaporation of hexadecane at higher temperatures. They could only be measured for 5 °C/min.

Samples SEBS-087-18-27-hex and SEBS-085-18-52-hex have no detectable hard-gel transition point, G’ decreases above the melting point of the paraffin (18 °C) without reaching a plateau. SEBS-118-12-26-hex shows fluid-like character immediately after the paraffin melted and without any hint of gel transition.

In the following, the influence of molecular weight, styrene content, and E/B ratio in the supporting SEBS on the thermo-mechanical properties is investigated in more details. The five samples of Table 1 have nearly the same styrene content of about 30% and the same ethylene/butylene ratio; therefore, this series is used to elucidate the effect of molecular weight. All the samples have similar G’ values when hexadecane is melted (Figure 5). They maintain this value over their specific hard-gel states. It is obvious that the transition temperatures from hard to soft gel shift to higher temperatures with increasing molecular weight of the supporting SEBS. Figure 6 depicts the transition temperatures *T*_m_, *T*_1_, and *T*_2_ over the molecular weight.

In this series, SEBS-345-28-64-hex with the highest molecular weight of 345 kg/mol reaches the highest value of *T*_1_ and *T*_2_ with 154 and 172 °C, respectively, and thus the highest shape stability. In contrast, SEBS-065-30-67-hex has the lowest molecular weight with 65 kg/mol and reaches its transition temperatures at considerably lower temperatures (51 and 52 °C). Both transition temperatures *T*_1_ and *T*_2_ show a degressive relationship between molecular weight and shape stability which is typically expected for mechanical properties of polymers [24]. The difference between *T*_1_ and *T*_2_ increases from 1 °C at SEBS-065-30-67-hex to 16 °C at SEBS-345-28-64-hex, indicating stronger physical cross-linking with increasing molecular weight. In the same way, the remaining two parameters, styrene content and E/B ratio, are investigated. SEBS-065-30-67-hex and SEBS-065-42-66-hex differ in the styrene content while both *M*_w_ and E/B ratio are comparable (Figure 7a). The shape stability increases from 51 °C at 30% styrene to 65 °C at 42% styrene. In addition, SEBS-080-29-64-hex and SEBS-085-29-64-hex with nearly same M_w_ also show an increase of shape stability with increasing styrene content. In both series, the impact of styrene content on thermo-mechanical properties is significant. A higher styrene content leads to larger extent physical cross-linking through the styrene domains, enhancing the shape stability at higher temperatures.

The influence of the ethylene content in the E/B phase can be seen in Figure 7b. Here, SEBS with 27% ethylene in the E/B phase (SEBS-087-18-27-hex) is compared to SEBS with 52% ethylene in E/B phase (SEBS-085-18-52-hex) while both, molecular weight and styrene content are comparable. Both samples turn directly into the soft gel state and do not show an intermediate hard gel state. Their gel–sol transition temperatures are at 42 and 43 °C, respectively. Therefore, the ethylene content in the E/B phase has no significant influence on the thermo-mechanical properties of FSPCMs at lower molecular weight. Obviously, only the molecular parameters within the styrene phase impact the shape stability of FSPCM, e.g., by acting as physical cross-linking sites. To correlate both impact factors—molecular weight and styrene content—the molecular weight of styrene block length is calculated according to Equation (1). The results are summarized in Figure 8.

Equation (1): Molecular weight of styrene block in SEBS
(1)$$Mw\;\left(styrene\;block\right)=\frac{Mw\;\left(SEBS\right)\times styrene\;content}{2}$$

All SEBS/hexadecane composites exhibit the same dependency: Increasing molecular weight of the styrene block leads to higher transition points *T*_1_ and *T*_2_ while *T*_m_ remains constant. With decreasing styrene block chain length, the soft gel area (range between *T*_1_ and *T*_2_) decreases slightly and disappears between *M*_w_ of 9.7 and 7.8 kg/mol. Finally, the thermo-mechanical properties of hard gel–soft gel transition can be attributed to the molecular weight of styrene block, which acts as physical crosslinking points. Et al Zhang [8] supposed that *T*_1_ might be related to the glass transition temperature *T*_g_ of the PS end blocks. It is further well-known that *T*_g_ of PS correlates with degree of polymerization *X* of styrene block in SEBS, Equation (2) [24,25].

Equation (2): Glass transition point on degree of polymerization for PS
(2)$$Tg\left(X\right)=\frac{Tg\left(\infty \right)}{1+2.4\times \left(\frac{1}{X}\right)}$$

It is obvious that *T*_1_ of SEBS-345-28-64-hex with 161 °C is appreciably higher than the glass transition point *T*_g_ of polystyrene phase with 104 °C and *T*_1_ of sample SEBS-065-30-67 is with 51 °C much lower than *T*_g_ of the polystyrene phase with lower degree of polymerization of 89 °C. Thus, the transition from gel to fluid cannot only be attributed to T_g_ of SEBS end blocks as supposed by Zhang et al. [8] where *T*_1_ and *T*_g_ are coincidentally the same comparing sample SEBS-138-31-65 [24,25,26]. Even though paraffin is mainly in the E/B phase, it may penetrate into the PS phase and dissolve it [8,21,27]. A lower molecular weight of the PS phase means shorter chains in the end blocks and thus promotes the dissolution process as known from the Flory–Huggins model [28,29,30]. The influence of the E/B to paraffin ratio can be seen by comparing samples SEBS-087-18-27 and SEBS-085-18-52. A difference from 27% to 52% ethylene in E/B phase has no influence on the transition temperatures *T*_1_ and *T*_2_. The higher styrene content the lower the E/B content and paraffin is forced to enter styrene domain at lower temperatures as expected. Therefore, the sample SEBS-065-41-66 has lower *T*_1_ and *T*_2_ values as expected, which may be traced back to the high styrene content. Comparing these results with the evaluation of the DSC analysis, it is obvious that all samples with *T*_1_ around 50 °C or below have a narrow peak. During DSC measurements, the samples pass through the soft-gel area, get viscous and finally diffluent. This change of geometrical dimensions leads to a faster energy input into the sample and thus to more narrow peaks in the second heating phase.

### 3.4. Leakage Behavior of Hexadecane in FSPCM

The leakage behavior was evaluated by determining the relative mass loss of hexadecane at 60 °C for the following samples in which *T*_1_ is well above the storage temperature: SEBS-345-28-64-hex, SEBS-194-31-64-hex, and SEBS -138-31-65. The time-dependent course of mass loss is shown in Figure 9. All three curves show degressive behavior that can be described by the Korsmeyer–Peppas equation, which is a special case of the Higuchi equation, Equation (3), and represent the diffusion of small molecules through a polymer [31,32]:

Equation (3): Korsmeyer–Peppas equation
(3)$$\frac{M_{t}}{M_{\infty }}=kt^{n}$$

*M_t_* is the amount of PCM lost at time t. *M_∞_* is amount of PCM totally stored in the sample at *t* = 0. The two parameters *k* and *n*, as seen in Table 6, describe the leakage behavior as follows. Structural and geometric characteristics of the supporting material are expressed by the kinetic release constant *k*. The exponent *n* provides information about the diffusion mechanism and represents the transport mechanism of the hexadecane through the matrix. The exponent *n* describe the amount of *M_t_/M_∞_* and should be below 0.6 [33].

The exponent *n* provides information about the diffusion mechanism and can be distinguished in several cases. If *n* = 0.5, the ratio *M_t_/M_∞_* is proportional to the square root of time; this case is called Fickian diffusion (Case I) and is often observed in polymers when the temperature is well above the glass temperature (T_g_) of the polymer. Although, properly, the process is pseudo-Fickian with *n* < 0.5, it is common to combine these two cases and consider Fickian diffusion when *n* < 0.5 as in our measurements [22,32]. SEBS-138-31-65 and SEBS-194-31-64 have both the same styrene and ethylene content in the E/B phase, but differ in the *M*_w_. The paraffin leakage between these two samples is in the same range suggesting that the increase in molecular weight from 138 kg/mol to 194 kg/mol does not affect the leakage behavior significantly. With further rising *M*_w_ to 345 kg/mol (SEBS-345-28-64-hex), the amount of mass loss is reduced by more than 7% at the same storage time compared to SEBS-194-31-64-hex and SEBS-138-31-65-hex. This may be ascribed to the increasing size of E/B domain where the paraffin is located preferentially. Just as Lee et al. [34] show that with increasing molecular weight of HDPE as a matrix for paraffin, the leakage behavior decreases, the leakage behavior of the SEBS system can be minimized with increasing molecular weight [34].

## 4. Conclusions

In this work, the influence of structure and chemical composition of SEBS as supporting material in SEBS/hexadecane form-stable PCM on the thermo-mechanical properties and leakage behavior were investigated. The thermal properties were investigated by TGA and DSC, thermo-mechanical properties were analyzed by temperature dependent rheological measurements and the leakage behavior was studied by weight loss in an oven. The most important influence on thermo-mechanical properties is the molecular weight of the styrene block. With increasing molecular weight of the styrene block in the supporting material, a declining increase of the thermo-mechanical properties for the SEBS/hexadecane composite is observed. The transition from gel-like to viscous behavior is not a softening by exceeding T_g_ of polystyrene, but a solution process where the physical crosslinking points are dissolved by paraffin. The highest retention behavior in the leakage test was recorded for the SEBS with the highest M_w_ SEBS-345-28-64. SEBS-194-31-64 and SEBS-138-31-65 with lower *M*_w_ show the same retention over time. This work creates an amplified understanding in shape stabilization of hexadecane with SEBS as matrix material with respect to structural parameters influencing the melting enthalpy, leakage behavior, and the thermo-mechanical properties. Future potential for reducing the leakage of hexadecane/paraffin from the SEBS/hexadecane FSPCM lies in the structural optimization of the supporting SEBS. Based on the results of this work, new SEBS types can be synthesized and the styrene block length can be adjusted to the desired temperature stability.

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
