# Peer review of "Form-Stable Phase Change Materials Based on SEBS and Paraffin: Influence of Molecular Parameters of Styrene-b-(Ethylene-co-Butylene)-b-Styrene on Shape Stability and Retention Behavior"

_materials, 2020, doi:10.3390/ma13153285_

Round 1
Reviewer 1 Report
The authors have developed and characterized phase change materials (PCM) based on styrene-b-(ethylene-co-butylene)-b-styrene / hexadecane system. They studied the influence of structure and chemical composition of PCMs, performing TGA, DSC and rheological measurements and also, investigating leakage behavior of PCMs. Finally, the authors showed that the most important influence on thermo-mechanical properties is the molecular weight of the styrene block. Based on the results of their work, new SEBS/hexadecane composites can be synthesize and the styrene block length can be adjusted to the desired temperature stability.
Some minor revisions are necessary:
1. At page 8, line 241: The authors should remove the first "samples" word (i.e. "The samples five samples").
2. At page 10, line 273 (Figure 7): Please add "a" and "b" labels on the images.
3. At page 10, line 273 (Figure 7a): The upper-left corner legend should be corrected for "Mw: 65 kg/mol" data (i.e. Tm, T1, T2, instead of T2, T2, T2).
4. At page 10, lines 291-292: The sentence "with increasing molecular weight of the styrene block, the transition points T1 and T2 behave degressively" looks like is not in concordance with Figure 6. Please reformulate.
5. At page 10, line 296: The name of first author of the reference [8] is missing.
6. At page 11, line 307: Same comment as previous.
Author Response
Dear Reviewer:
First off all, thanks for your Feedback. Below you will see our comments to your review in blue respectively. I hope we could correct it to your satisfaction.
1. At page 8, line 241: The authors should remove the first "samples" word (i.e. "The samples five samples").
At page 10, line 249: the word "samples" is removed
2. At page 10, line 273 (Figure 7): Please add "a" and "b" labels on the images.
At page 12, line 281 (Figure 7): the labels "a" and "b" are added in the images.
3. At page 10, line 273 (Figure 7a): The upper-left corner legend should be corrected for "Mw: 65 kg/mol" data (i.e. Tm, T1, T2, instead of T2, T2, T2).
At page 12, line 281 (Figure 7): the legend is corrected from T2, T2, T2 to Tm, T1, T2 in the Image 7a).
4. At page 10, lines 291-292: The sentence "with increasing molecular weight of the styrene block, the transition points T1 and T2 behave degressively" looks like is not in concordance with Figure 6. Please reformulate: .
At page 12, line 299-301: The sentence is refomulated from"with increasing molecular weight of the styrene block, the transition points T1 and T2 behave degressively" to "All SEBS/hexadecane composites exhibit the same dependency: Increasing molecular weight of the styrene block leads to higher transition points T1 and T2 while Tm remains constant".
5. At page 10, line 296: The name of first author of the reference [8] is missing.
At page 12, line 304: The Name of first author of the reference [8] is added
6. At page 11, line 307: Same comment as previous.
At page 13, line 316: The Name of first author of the reference [8] is also added
Many thanks in advance
Ralf Rickert
Reviewer 2 Report
The authors present results of testing properties of phase change materials (PCMs) based on styrene -b-(ethylene-co-butylene)-b-styrene (SEBS) and paraffin. In this regard, they focused on the experimental testing influence of molecular parameters of SEBS on thermo-mechanical properties and leakage behaviour.
The research aimed at testing the properties and possibilities of using PCMs is beneficial from an economic and ecological point of view. Thanks to their thermo-physical properties, PCMs can be used in several applications and provide significant energy savings.
In my opinion, the topic of the presented article is interesting and inspiring. However, the current version of the manuscript contains some bugs and inaccuracies that need to be corrected. Therefore, I propose a major revision, as follows:
Title: It is not suitable to use abbreviations.
Abstract: line 22: There should be …thermo-mechanical… instead of … thermos- mechanical
All abbreviations should be explained the first time they appear in the main text. For example:
Line 71: NMR, line 97 SDV, line 102: THF, etc.
Line 81: There should be 18°C instead of 18°.
Table 4: Units of T99%, T15%, ΔHc are missing.
Figure 3: I miss the legend describing the individual curves. I also miss the marking a), b) in the pictures.
The results presented in Figure 3 should be described in more detail and explained in the text of the paper.
Figure 7, 9: I miss the marking a), b) in the pictures.
Figure 9: Please check the axis of Fig.9a. Is the unit of ratio Mt/M∞ infinity (%) correct?
References: Please check all references style according to Guidelines for authors: https://www.mdpi.com/authors/layout#_bookmark36
Author Response
Dear Reviewer:
First off all, thanks for your Feedback. Below you will see our comments to your review in blue respectively. I hope we could correct it to your satisfaction.
Title: It is not suitable to use abbreviations.
Title: The Abbreviation "SEBS" is replaced to styrene-b-(ethylene-co-butylene)-b-styrene .
Abstract: line 22: There should be …thermo-mechanical… instead of … thermos- mechanical
Abstract: line 23: The word "thermos-mechanical" is replaced to "thermo-mechanical"
All abbreviations should be explained the first time they appear in the main text. For example:
Line 71: NMR, line 97 SDV, line 102: THF, etc.
Line 71 NMR is removed, the Abbreviation is explained in 2. aterials and Methods Line 103.
the explaination of SDV: styrene divinyl benzene (Line 95) and Tetrahydrofuran THF (Line 101) are explained respectively
Line 81: There should be 18°C instead of 18°.
Line 81: The unit C for celsius is added
Table 4: Units of T99%, T15%, ΔHc are missing.
The Units of T99%, T15% are added, the unit for ΔHc was there and the first line of Table 4 had to be pulled down.
Figure 3: I miss the legend describing the individual curves. I also miss the marking a), b) in the pictures.
Figure 3: The curves are named now. And the missing marking a) and b) are added.
The results presented in Figure 3 should be described in more detail and explained in the text of the paper
The results presented in Figure 3 are descibed in this passage (Line 196 -208)
The passage:
"Comparing these results with the evaluation of the DSC analysis, it is obvious that all samples with T1 around 50 °C or below have a narrow peak. During DSC measurements, the samples pass through the soft-gel area, get viscous and finally diffluent. This change of geometrical dimensions leads to a faster energy input into the sample and thus to more narrow peaks in the second heating phase."
Line 322-326 is also added and explain the DSC results.
Figure 7, 9: I miss the marking a), b) in the pictures.
The missing marks a) and b) are added to Figure 7 and 9.
Figure 9: Please check the axis of Fig.9a. Is the unit of ratio Mt/M∞ infinity (%) correct?
Figure 9: The axis Mt/M∞ is changed from 0.X to X %
References: Please check all references style according to Guidelines for authors: https://www.mdpi.com/authors/layout#_bookmark36
All References are changed. The order chaneged from alphabetical to numbered in the order cited in the paper.
Reviewer 3 Report
Paper by Dr. Frank Schönberger “Form-stable phase change materials based on SEBS and paraffin: Influence of molecular parameters of SEBS on shape stability and retention behavior” deals with study of the influence of molecular parameters of SEBS triblock copolymers on the thermo-mechanical properties and leakage behavior of their mixtures with paraffin. The manuscript consists of some new interesting results but revision is required. Some misprints should be corrected. For example, in abstract, line 22, “thermos- mechanical” should be changed to “thermomechanical”. In table 1, some important numbers are lost.
Author Response
Dear Reviewer:
First off all, thanks for your Feedback. Below you will see our comments to your review in blue respectively. I hope we could correct it to your satisfaction.
For example, in abstract, line 22, “thermos- mechanical” should be changed to “thermomechanical”
In Abstract, line 23: "thermos- mechanical" is changed to "thermo- mechanical"
In table 1, some important numbers are lost:
The missing numbers in Table 1 were based on a too narrow culomn. The table was fixed.
Many thanks in advance
Ralf Rickert
Round 2
Reviewer 2 Report
The authors accepted my comments and suggestions.